# Genome-Wide Identification, Molecular Characterization, and Expression Analysis of the HSP70 and HSP90 Gene Families in *Thamnaconus septentrionalis*

**DOI:** 10.3390/ijms25115706

**Published:** 2024-05-24

**Authors:** Ying Chen, Qing Chang, Qinmei Fang, Ziyang Zhang, Dan Wu, Li Bian, Siqing Chen

**Affiliations:** 1College of Fisheries and Life Sciences, Shanghai Ocean University, Shanghai 201306, China; 13155031583@163.com; 2State Key Laboratory of Mariculture Biobreeding and Sustainable Goods, Yellow Sea Fisheries Research Institute, Chinese Academy of Fishery Sciences, Qingdao 266071, China; changqing@ysfri.ac.cn (Q.C.); 18330368260@163.com (Z.Z.); wd_wudan@163.com (D.W.); 3Fujian Academy of Agricultural Sciences, Fuzhou 350003, China; faasfang@126.com

**Keywords:** *Thamnaconus septentrionalis*, HSP70, HSP90, tissue, developmental stage, abiotic stress

## Abstract

Heat shock proteins (HSPs) are a class of highly conserved proteins that play an important role in biological responses to various environmental stresses. The mariculture of *Thamnaconus septentrionalis*, a burgeoning aquaculture species in China, frequently encounters stressors such as extreme temperatures, salinity variations, and elevated ammonia levels. However, systematic identification and analysis of the HSP70 and HSP90 gene families in *T. septentrionalis* remain unexplored. This study conducted the first genome-wide identification of 12 HSP70 and 4 HSP90 genes in *T. septentrionalis*, followed by a comprehensive analysis including phylogenetics, gene structure, conserved domains, chromosomal localization, and expression profiling. Expression analysis from RNA-seq data across various tissues and developmental stages revealed predominant expression in muscle, spleen, and liver, with the highest expression found during the tailbud stage, followed by the gastrula, neurula, and juvenile stages. Under abiotic stress, most HSP70 and HSP90 genes were upregulated in response to high temperature, high salinity, and low salinity, notably *hspa5* during thermal stress, *hspa14* in high salinity, and *hsp90ab1* under low salinity conditions. Ammonia stress led to a predominance of downregulated HSP genes in the liver, particularly *hspa2*, while upregulation was observed in the gills, especially for *hsp90b1*. Quantitative real-time PCR analysis corroborated the expression levels under environmental stresses, validating their involvement in stress responses. This investigation provides insights into the molecular mechanisms of HSP70 and HSP90 in *T. septentrionalis* under stress, offering valuable information for future functional studies of HSPs in teleost evolution, optimizing aquaculture techniques, and developing stress-resistant strains.

## 1. Introduction

Heat shock proteins (HSPs), also known as stress proteins, are synthesized in organisms in response to environmental stress, enhancing cellular resilience to extreme conditions [1]. Furthermore, HSPs serve as highly conserved molecular chaperones, facilitating diverse physiological functions through binding and assisting in the intracellular folding, assembly, transportation, or degradation of peptides [2]. Originally identified in *Drosophila melanogaster*, subsequent studies have confirmed their widespread presence across the biosphere, from bacteria to humans [3]. Structurally, HSPs are categorized into six major classes based on molecular weight: HSP110, HSP90, HSP70, HSP60, HSP40, and small heat shock protein (sHSP) [4]. For aquatic animals within the vast HSP family, the HSP70 and HSP90 gene families have emerged as focal points of research, given their pivotal roles in modulating stress responses [5,6,7].

Aquatic species, particularly fish, are inherently vulnerable to a spectrum of external environmental pressures [8,9], including thermal extremes, osmotic stress, and ammonia toxicity. The expansion of whole-genome sequencing across various species has paved the way for in-depth studies on genes implicated in environmental stress adaptation, with HSP70 and HSP90 proteins drawing considerable attention. These proteins have been extensively identified, analyzed, and linked to stress response mechanisms in teleosts. For example, in the *Lateolabrax maculatus*, investigations identified 5 HSP90 genes and 16 HSP70 genes, with their expression significantly upregulated in gills, liver, and muscle following thermal stress, suggesting their critical role in heat stress adaptation [10]. Similarly, in *Sebastiscus marmoratus*, 15 HSP70 genes were identified, with a majority showing upregulation under thermal stress [11]. Studies have also highlighted the differential expression of HSP70 genes in *Boleophthalmus pectinirostris* under ammonia stress [12], as well as the distinct expression patterns of HSP70 and HSP90 genes under osmotic and hypoxic stress [13,14]. Notably, in *Scophthalmus maximus*, the differential expression of identified HSP70 genes under both biotic and abiotic stresses was verified [2]. The above studies illustrate the regulatory functions of HSP70 and HSP90 genes in stress responses and reveal their species-specific functional discrepancy [10].

*Thamnaconus septentrionalis*, classified within the Tetraodontiformes, Monacanthidae, and *Thamnaconus*, inhabits offshore demersal zones at depths of 50 to 120 m and is predominantly distributed around the Korean Peninsula, Japan, and Chinese maritime regions [15]. This species also represents a traditional fishery target, ranking as the second-largest conventional fishing catch in China during the 1970s, following *Trichiurus japonicus* [16]. Since the 1990s, excessive fishing and environmental degradation have significantly reduced the catch volumes of *T. septentrionalis*, failing to meet market demand [17]. Moreover, its dense flesh, absence of intermuscular bones, and rich nutritional content have made it highly favored by consumers [18], leading to an expansion in aquaculture. *T. septentrionalis* thrives in salinity levels of 25 to 35 and temperatures of 21 to 24 °C [19]. Cage culture, primarily in Fujian Province, China, is the predominant cultivation method for this species. Summer brings a period of high temperatures lasting 1 to 2 months, while the significant river runoff during the rainy season may impede seawater exchange, leading to abrupt changes in salinity. Additionally, excessive aquaculture density might elevate the concentration of ammonia nitrogen in the water. Consequently, *T. septentrionalis* in aquaculture settings is susceptible to various environmental stresses, including temperature, salinity, and ammonia nitrogen. Therefore, investigating the HSP70 and HSP90 genes at the molecular level to uncover their molecular response mechanisms under environmental stress conditions holds substantial practical significance for guiding the optimization of aquaculture techniques for *T. septentrionalis* in adverse environments.

Upon the completion of whole-genome sequencing and assembly for *T. septentrionalis* [15], our investigation embarked on extensive genomic scrutiny of HSP70 and HSP90 gene families. The research began with an examination of the essential molecular features, covering the proteins’ physicochemical characteristics and subcellular localization. Following this, the evolutionary lineage of these genes was explored through a thorough phylogenetic analysis, gene structure analysis, conserved domain and motif analyses, and chromosomal mapping. To further comprehend the expression profiles of HSP70 and HSP90 genes within *T. septentrionalis* across a range of tissues, developmental stages, and under various environmental stresses, we analyzed RNA-seq datasets. These datasets encompassed seven distinct tissues (heart, brain, liver, spleen, gill, intestine, and muscle), eight developmental stages (cleavage (Cle), blastocyst (Bla), gastrula stage (Gas), neural embryonic stage (Neu), tail bud stage (Tai), muscle effect stage (Mus), hatchling (Dph0) and 20 days of juvenile (Dph20)), and four environmental stresses, including high temperature, high salt, low salt and ammonia nitrogen stresses. The qPCR analyses of the HSP70 and HSP90 genes under these stress conditions reinforce their expression trends under environmental challenges. This comprehensive approach not only contributes valuable resources to the field of heat shock protein research in teleost but also provides a foundational understanding of the HSPs’ molecular mechanisms under different environmental stresses in *T. septentrionalis*.

## 2. Results

### 2.1. Identification and Characteristics of HSP70 and HSP90 Genes in T. septentrionalis

Utilizing BLASTp alignment and HMM search, this study identified a total of 12 HSP70 genes and 4 HSP90 genes within the genome of *T*. *septentrionalis*. Following the nomenclature convention for heat shock proteins in osteichthyes, the members of the HSP70 gene family in *T*. *septentrionalis* were designated as *hsp70*, *hspa1b*, *hspa4*, *hspa4a*, *hspa4l*, *hspa5*, *hspa8.1*, *hspa8.2*, *hspa9*, *hspa13*, *hspa14*, and *hyou1*. The HSP90 gene family members were named *hsp90aa1*, *hsp90ab1*, *hsp90b1*, and *trap1*. Gene accession numbers for HSP70 and HSP90 in teleosts and other animals were detailed in Appendix A, while a comparison of the HSP70 and HSP90 gene counts between *T*. *septentrionalis* and other species was presented in Appendix A.

Utilizing the ExPASy online platform, this study conducted a physicochemical property analysis of HSP70 and HSP90 proteins in *T*. *septentrionalis*. Details such as gene names and protein physicochemical properties of each gene family member were presented in Table 1. The amino acid lengths of HSP70 proteins ranged from 442 (*hspa13*) to 945 (*hyou1*), with molecular weights varying between 47.83 (*hspa13*) and 106.41 kDa (*hyou1*) and isoelectric points between 4.97 (*hspa5*) and 6.24 (*hspa9*). HSP90 proteins displayed amino acid lengths from 719 (*trap1*) to 801 (*hsp90b1*), molecular weights from 81.84 (*trap1*) to 91.99 kDa (*hsp90b1*), and isoelectric points from 4.74 (*hsp90b1*) to 5.99 (*trap1*). The protein instability index results indicated that out of 16 genes, 9 were classified as stable proteins (instability index < 40), while 7 were identified as unstable (instability index > 40); the aliphatic index ranged from 72.97 (*hspa4*) to 102.15 (*hspa13*). Hydrophobicity analysis of HSP70 and HSP90 proteins revealed that apart from *hspa13* exhibiting certain hydrophobicity (>0), other HSPs displayed hydrophilic characteristics (<0). All sequences of HSP70 and HSP90 in *T*. *septentrionalis* had been submitted to the GenBank database, and their NCBI accession numbers were listed in Table 1.

### 2.2. Phylogenetic Analysis of HSP70 and HSP90 Genes

A phylogenetic tree of the HSP70 gene in *T. septentrionalis* was constructed based on the identified HSP70 protein sequences in *Danio rerio*, *Oryzias latipes*, *Gasterosteus aculeatus*, *Larimichthys crocea*, *Oreochromis niloticus*, *Homo sapiens*, *Mus musculus*, and *Gallus gallus*. As illustrated in Figure 1, the phylogenetic tree of HSP70 was divided into three major branches, with the 12 HSP70 genes categorized into eight clusters: hspa1, hspa4, hspa5, hspa8, hspa9, hspa13, hspa14, and hyou1. Gene duplication events were observed within the branches of hspa1 (*hsp70*, *hspa1b*), hspa4 (*hspa4*, *hspa4a*, and *hspa4l*), and hspa8 (*hspa8.1* and *hspa8.2*), while no gene duplication was noted in the branches of hspa5, hspa9, hspa13, hspa14, and hyou1. The 12 genes were grouped with orthologous genes from different species within the eight branches, exhibiting high bootstrap values and indicating robust phylogenetic relationships.

A phylogenetic tree for the HSP90 gene in *T. septentrionalis* was constructed using identified HSP90 protein sequences from *D. rerio*, *O. latipes*, *L. crocea*, *O. niloticus*, *Salmo salar*, *Ictalurus punctatus*, *Lepisosteus oculatus*, *H. sapiens*, *M. musculus*, *G. gallus*. As depicted in Figure 2, the phylogenetic tree of HSP90 was divided into three major branches, with the 4 HSP90 genes categorized into four distinct branches: hsp90aa1, hsp90ab1, hsp90b1, and trap1. No gene duplication events were observed within any of the branches. The protein sequences from all species used in constructing the evolutionary trees are available in Appendix A.

### 2.3. Gene Structure and Chromosome Distribution Analysis

The gene structures of HSP70 and HSP90 genes in *T. septentrionalis* were analyzed using TBtools software (version 2.092, Dr. Chengjie Chen, Guangdong, China) [20]. As illustrated in Figure 3, except for *hspa1b* and *hsp70*, which contained only one coding sequence (CDS), all other HSP70 and HSP90 genes possessed more than four CDSs. Furthermore, when gene structure was examined in conjunction with phylogenetic relationships using TBtools, genes with homologous relationships exhibited similar gene structures, such as *hspa8.1* and *hspa8.2*, as well as *hsp70* and *hspa1b*. The lengths of HSP70 genes ranged from 2 to 18 kb, while those of HSP90 genes span from 4 to 11 kb.

Chromosomal localization of HSP70 and HSP90 genes in *T. septentrionalis* was also conducted using TBtools software (version 2.092, Dr. Chengjie Chen, Guangdong, China), as shown in Figure 4. The 12 HSP70 genes were distributed across nine chromosomes, with three genes (*hspa4a*, *hyou1*, and *hspa8.1*) located on chromosome 4 and two genes (*hspa4* and *hspa9*) on chromosome 9. The remaining HSP70 genes were individually located on chromosomes 2, 10, 11, 12, 15, 17, and 18. Among the HSP90 genes, *hsp90aa1* and *hsp90ab1* were found on chromosome 19, *hsp90b1* was on chromosome 13, and *trap1* was located on chromosome 15.

### 2.4. Motif and Conserved Domain Analysis

Conserved motif analysis of HSP70 and HSP90 protein sequences in *T. septentrionalis* was conducted using the online MEME website, with the number of motifs set to 20. As shown in Figure 5, the detected number of motifs for HSP70 genes ranged from 8 to 14, with the majority of HSP70 genes sharing nine common motifs: motifs 1, 2, 3, 4, 6, 7, 9, 10, and 11. All HSP70 genes shared seven motifs, specifically motifs 1, 3, 4, 6, 7, 9, and 11. The HSP90 genes exhibited a range of 5 to 9 detected motifs, with all HSP90 genes sharing five common motifs: motifs 12, 13, 14, 17, and 18. Overall, the variety of motifs across different genes was relatively consistent, and genes with homologous relationships displayed identical motif patterns, for example, *hsp70*, *hspa8.2*, *hspa1b*, and *hspa5*; *hspa4*, *hspa4l*, and *hspa4a*.

Conserved Domain Analysis of HSP70 and HSP90 genes in *T. septentrionalis* was performed using the NCBI website’s Batch CD-Search tool. Subsequently, TBtools software (version 2.092, Dr. Chengjie Chen, Guangdong, China) was employed to integrate the phylogenetic tree, motifs, and conserved domains. As illustrated in Figure 5, among the 12 HSP70 genes, five possessed the PTZ00009 superfamily domain (*hspa8.1*, *hspa8.2*, *hspa1b*, *hsp70*, and *hspa5*), three genes were characterized by the NBD_sugar-kinase_HSP70_actin superfamily domain (*hspa4a*, *hspa4*, and *hspa14*), and the remaining domains included dnak (*hspa9*), HSPA13-like_NBD (*hspa13*), HYOU1-like_NBD (*hyou1*), and HSPA4_like_NBD (*hspa4l*). Furthermore, among the four HSP90 genes, two had the PRK14083 superfamily domain (*hsp90aa1* and *hsp90ab1*), the *trap1* gene contained the PRK05218 domain, and the *hsp90b1* gene encompassed both the HATPase_HSP90-like domain and the HSP90 domain. A phylogenetic tree was constructed using the HSP70 and HSP90 protein sequences of *T. septentrionalis*, and TBtools software (version 2.092, Dr. Chengjie Chen, Guangdong, China) facilitated the collective visualization of the phylogenetic tree, motifs, and conserved domains.

### 2.5. Protein Structure Prediction and Subcellular Localization

As depicted in Table 2 and Figure 6, the secondary structures of HSP70 proteins in *T. septentrionalis* predominantly comprised alpha helices and random coils, with alpha helices accounting for 33.79% (*hspa8.1*) to 48.99% (*hyou1*), Extended Strand ranged from 13.23% (*hyou1*) to 24.90% (*hspa14*), Beta Turn from 3.07% (*hspa4*) to 8.54% (*hspa8.1*), and Random coil from 28.34% (*hspa9*) to 40.19% (*hspa4*). The secondary structures of HSP90 proteins in *T. septentrionalis* were mainly composed of alpha helices and random coils as well, with alpha helices representing 45.20% (*trap1*) to 54.68% (*hsp90b1*), Extended Strand from 12.36% (*hsp90b1*) to 13.77% (*trap1*), Beta Turn from 3.75% (*hsp90b1*) to 5.70% (*trap1*), and Random coil from 28.41% (*hsp90aa1*) to 35.33% (*trap1*). Appendix A presented the results of assessing the quality of the three-dimensional structure of HSP70 and HSP90 proteins and included the three-dimensional structural quality parameters in Table 2. The value of residues in the most favored regions [A, B, L] is above 90%, indicating that the constructed 3D models of the proteins were of high quality.

Subcellular localization of HSP70 and HSP90 genes in *T. septentrionalis* was conducted using the online website WOLF PSORT. Predictive results of subcellular localization revealed that 7 out of the 12 HSP70 genes (*hspa1b*, *hspa4*, *hspa4a*, *hspa4l*, *hspa8.1*, *hspa8.2*, and *hspa14*) were expressed in the Cytosol, 3 genes (*hspa5*, *hspa13*, and *hyou1*) in the endoplasmic reticulum, *hsp70* in the Nucleus, and *hspa9* in the mitochondrion. Among the HSP90 genes, *hsp90aa1* and *hsp90ab1* were expressed in the Cytosol, *hsp90b1* in the endoplasmic reticulum, and *trap1* in the mitochondrion. In summary, HSP70 and HSP90 genes were predominantly expressed in the cytoplasm, endoplasmic reticulum, and mitochondria.

### 2.6. Signal Peptide Prediction, Transmembrane Structural Domain Prediction and Selection Test on Duplicated HSP70 and HSP90 Gene Pairs

The signal peptide prediction results for HSP70 and HSP90 proteins indicated (Figure 7) that *hspa5* and *hyou1* from HSP70 and *hsp90b1* from HSP90 had been predicted and preliminarily identified as secretory proteins (SP). As shown in Figure 7, the highest cleavage site (CS) score for *hspa5* is 0.7102, occurring at the 16th position alanine (A); for *hyou1*, the highest CS score is 0.5765, found at the 23rd position threonine (T), and for *hsp90b1*, the highest CS score is 0.9034, at the 21st position alanine (A). Based on the SP values, proteins *hspa5*, *hyou1*, and *hsp90b1* were all equipped with signal peptides, and the lengths of these signal peptides were approximately 15, 22, and 20, respectively.

The transmembrane domain prediction results (Figure 8) indicated that among the 16 HSP proteins, only the HSP70 family member *hspa13* was likely to possess a transmembrane domain, suggesting that *hspa13* may function as a membrane protein. In summary, based on the prediction outcomes, none of the 16 HSP proteins exhibited both signal peptides and transmembrane domains.

To investigate the evolutionary constraints and selection pressures on HSP70 and HSP90 genes, TBtools software (version 2.092, Dr. Chengjie Chen, Guangdong, China) was utilized to calculate the nonsynonymous (Ka), synonymous (Ks), and Ka/Ks ratios for five homologous pairs of HSP70 genes and one pair of HSP90 genes. As indicated in Table 3, the Ka/Ks ratios for the HSP70 gene pairs ranged between 0.045851 and 0.182385, while the Ka/Ks ratio for the HSP90 gene pair was 0.115141. The Ka/Ks ratios for all six gene pairs were much below 1.0, suggesting that these genes had undergone strong purifying selection throughout their evolutionary history.

### 2.7. Expression Patterns of HSP70 and HSP90 Genes in Tissues of T. septentrionalis

To determine the functional expression of HSP70 and HSP90 genes across different tissues in *T. septentrionalis*, transcriptome results from various tissues of the species were analyzed, with FPKM values processed using TBtools software (version 2.092, Dr. Chengjie Chen, Guangdong, China) to study the expression profiles of HSP70 and HSP90 genes. As depicted in Figure 9, all 12 HSP70 and 4 HSP90 genes were expressed across seven tissues. In the liver, the expression of *hspa8.1* was the highest, while *hspa1b* was the lowest. In both the brain and heart, *hsp90ab1* exhibited the highest expression and *hspa8.2* the lowest. In the intestine, *hspa8.1* showed the highest expression, with *hspa4l* being the lowest. In muscle, *hspa8.2* was the highest, and *hspa13* was the lowest. In gills, *hsp90ab1* had the highest expression, and *hspa1b* was the lowest. In the spleen, *hspa8.1* was the highest, and *hspa8.2* was the lowest. Overall, the three genes with the highest expression across the seven tissues were *hspa8.1* (liver, intestine, and spleen), *hsp90ab1* (brain, heart, and gills), and *hspa8.2* (muscle), whereas the four genes with the lowest expression were *hspa8.2* (brain, heart, and spleen), *hspa1b* (liver and gills), *hspa4l* (intestine), and *hspa13* (muscle).

### 2.8. Expression Patterns of HSP70 and HSP90 Genes during Different Developmental Stages of T. septentrionalis

To ascertain the expression roles of HSP70 and HSP90 genes across different developmental stages in *T. septentrionalis*, transcriptome data from various developmental stages of the species were analyzed, with FPKM values processed using TBtools software (version 2.092, Dr. Chengjie Chen, Guangdong, China) to explore the expression profiles of HSP70 and HSP90 genes. As shown in Figure 10, all 12 HSP70 and 4 HSP90 genes were expressed across eight developmental stages. During the cleavage, blastocyst, neuroembryonic, and tail bud stages, *hsp90ab1* exhibited the highest expression, while *hspa8.2* showed the lowest. Throughout the gastrula stage, muscle effect stage, hatch 0-day larvae, and hatch 20-day juvenile, *hsp90ab1* again had the highest expression, with *hspa1b* displaying the lowest. In summary, the *hsp90ab1* gene consistently showed the highest expression across all eight developmental stages, while the two genes with the lowest expression throughout these stages were *hspa8.2* and *hspa1b*.

### 2.9. Expression Patterns of HSP70 and HSP90 Genes in Abiotic Stresses of T. septentrionalis

To determine the expression roles of HSP70 and HSP90 genes in *T. septentrionalis* under various abiotic stress conditions, transcriptome data from different environmental stresses were analyzed, with FPKM values processed using TBtools software (version 2.092, Dr. Chengjie Chen, Guangdong, China) to investigate the expression profiles of HSP70 and HSP90 genes. As shown in Figure 11, In the liver, compared to the control group, under high-temperature stress, all genes except *hspa8.2* were upregulated. Under high salinity stress, 11 genes were upregulated (*hspa4*, *hspa4a*, *hspa4l*, *hspa5*, *hspa8.1*, *hspa9*, *hspa13*, *hspa14*, *hyou1*, *hsp90ab1*, and *hsp90b1*), while 5 genes were downregulated (*hsp70*, *hspa1b*, *hspa8.2*, *hsp90aa1*, and *trap1*). Under low salinity stress, 10 genes were upregulated (*hspa4*, *hspa5*, *hspa8.1*, *hspa8.2*, *hspa9*, *hspa13*, *hspa14*, *hyou1*, *hsp90ab1*, and *hsp90b1*), with 6 genes downregulated (*hsp70*, *hspa1b*, *hspa4a*, *hspa4l*, *hsp90aa1*, and *trap1*). Under ammonia stress, 12 genes were upregulated, with 4 genes downregulated (*hspa1b*, *hspa4a*, *hspa4l*, and *hsp90aa1*). In the gills, compared to the control group, under ammonia stress, 10 genes were upregulated, while 6 genes were downregulated (*hsp70*, *hspa1b*, *hspa4l*, *hspa8.1*, *hspa8.2*, and *hsp90ab1*). In summary, under high temperature, high salinity, low salinity, and ammonia stress, seven genes were consistently upregulated, namely *hspa4*, *hspa5*, *hspa9*, *hspa13*, *hspa14*, *hyou1*, and *hsp90b1*, with *hspa1b* being consistently downregulated.

### 2.10. qPCR Validation of HSP70 and HSP90 Genes Expression Patterns under Abiotic Stresses

To validate the accuracy of the transcriptome results, all 12 HSP70 and 4 HSP90 genes were subjected to qPCR under conditions of high temperature, high salinity, and low salinity stress. As depicted in Figure 12, the qPCR results largely mirrored the trends observed in the RNA-seq data. This concordance indicated the reliability of the transcriptomic data obtained in this study.

## 3. Discussion

Changes in the external environment can influence the physiological and metabolic activities of fish [8]. At the molecular level, environmental stress can damage proteins, affecting their normal folding. Cellular responses to stress involve specific signaling events and the activation and extensive synthesis of molecular chaperones, many of which are known as heat shock proteins [21]. Heat shock proteins (HSPs) are a highly conserved class of chaperone proteins that can bind to numerous protein molecules, playing multiple physiological roles such as aiding in the correct folding of amino acid chains into their three-dimensional structures, repairing damaged proteins, and degrading misfolded proteins [22]. HSP70 and HSP90 proteins are ubiquitously present in bacteria to mammals, playing crucial roles in maintaining proteostasis, cellular differentiation and development, and protein conformational regulation [23,24,25]. Although recent years have seen increased research on fish HSP70 and HSP90 proteins, a comprehensive genomic identification of the HSP70 and HSP90 gene families in *T. septentrionalis* has not yet been reported. This study conducted a comprehensive analysis of the HSP70 and HSP90 gene families in *T. septentrionalis*, providing valuable resources for studying the roles of HSP gene families in stress resistance and fish evolution.

In this study, a total of 12 HSP70 and 4 HSP90 genes were identified in *T. septentrionalis*. The number of identified HSP70 and HSP90 genes was relatively similar to those found in humans [26] and other teleosts [27,28]. Compared to humans, the gene repertoire lacked *hspa1a*, *hspa1l*, *hspa2*, *hspa6*, *hspa7*, *hspa12a*, *hspa12b*, and *hsph1* genes, among which *hspa1l*, *hspa2*, *hspa6*, *hspa7*, and *hsph1* are also absent in most teleosts. This reflects the high degree of conservation of HSP70 genes throughout the evolution of teleost fish. Moreover, the absence of *hspa2*, *hspa6*, and *hspa7* genes in teleost fish, but their presence in mammals, suggests these genes likely evolved in higher vertebrates [29,30]. Additionally, except for zebrafish, the *hsph1* gene is missing in most teleost genomes, implying a potential loss of the *hsph1* gene during evolutionary processes. Similar to most teleosts [10,11], *T. septentrionalis* exhibits gene duplications within the *hspa4* (*hspa4*, *hspa4a*, and *hspa4l*) and *hspa8* (*hspa8.1* and *hspa8.2*) genes. The HSP90 gene family members in *T. septentrionalis* showed no differences compared to other species [10], indicating a high degree of conservation of HSP90 genes in teleost evolution. In summary, despite some variations in the number and members of HSP70 and HSP90 genes among teleost fish, the overall similarity underscores the high degree of conservation of HSP70 and HSP90 genes throughout teleost evolution.

Subcellular localization indicated that (Table 2), among the 12 HSP70 genes, 7 were located in the cytosol, 3 in the endoplasmic reticulum (ER), with the remaining two distributed between the mitochondrion and the nucleus; of the 4 HSP90 genes, 2 were situated in the cytosol, 1 in the ER, and 1 in the mitochondrion. These findings align with subcellular localization results for *S. maximus* [2] and *S. marmoratus* [11], where the majority of genes were located in the cytosol, with the rest in the ER, mitochondrion, and nucleus. This suggests a higher functional engagement of HSP genes within the cytosol. Studies have demonstrated the critical role of cytosolic HSP70 genes under stress conditions [31], and the diversity in subcellular localization may correlate with the varied functions of HSP genes [32].

Signal peptide predictions revealed that HSP70′s *hspa5* and *hyou1* and HSP90′s *hsp90b1* proteins possessed signal peptides, indicating their presence in the ER as secretory proteins. The proteins *hspa5*, *hyou1*, and *hsp90b1,* serving as ER chaperones, play crucial roles in the quality control of protein folding and synthesis within the ER [33]. Similar findings in *S. maximus* studies [2], where *hspa5* and *hyou1* proteins of the HSP70 gene family also had signal peptides, preliminarily identified as secretory proteins, reflected the high degree of evolutionary conservation within the HSP70 family.

Gene duplication serves as the foundation for gene family expansion, providing the primal material for the formation of new genes and the innovation of gene functions. It plays a crucial role in species evolution [34,35,36], achievable through whole genome duplication (WGD) or small-scale duplication (SSD) [37,38]. In this study, we observed duplicated genes within HSP70, including *hspa4* (*hspa4*, *hspa4a*, and *hspa4l*) and *hspa8* (*hspa8.1* and *hspa8.2*), while no duplicated genes were identified in HSP90. Chromosomal localization suggested that the duplicated genes *hspa4*, *hspa4a*, and *hspa4l*, as well as *hspa8.1* and *hspa8.2*, were located on different chromosomes, indicating the absence of tandem duplications and suggesting that the amplification of *T. septentrionalis* HSP70 gene likely occurred after the two rounds of WGD in early vertebrate evolution. However, the *hspa8a.1* and *hspa8a.2* of *S. maximus*, located on Chr 2, were identified as tandem duplications, and another *hspa8b* gene was located on chromosome 11, suggesting SSD events in the evolution of *S. maximus* posted the two rounds of WGD in early vertebrates [2,39]. Additionally, the Ka/Ks ratios of duplicated gene pairs within *T. septentrionalis* HSP70 and HSP90 genes were much less than 1, indicating that duplicated genes in *T. septentrionalis* had undergone strong purifying selection throughout their evolution. Similarly, the Ka/Ks ratios for duplicated gene pairs in *S. maximus* [2] and *S. marmoratus* [11] were also less than 1.

Gene structure and motif play crucial roles in the evolution of gene families and provide compelling evidence for their evolutionary trajectories [40]. Motif analysis revealed that the *hspa8.2*, *hspa1b*, *hsp70*, and *hspa5* genes shared identical motifs 1~11 as well as motifs 5 and 6, while *hspa4* and *hspa4l* possessed the same motifs 1~4, 6, 7, 9, 10, 11, 15, 16, and 20, with identical types and distributions of motifs. Gene structure diagrams indicated that *hspa1b* and *hsp70* have highly similar gene structure distributions, with variations observed in the structures of other genes. These results, combined with conserved domains and phylogenetic relationships, suggested an exceedingly close kinship between the *hspa1b* and *hsp70* genes. Moreover, all HSP70 genes shared seven motifs (motifs 1, 3, 4, 6, 7, 9, and 11), and all HSP90 genes possessed five motifs (motifs 12, 13, 14, 17, 18). Studies have demonstrated that gene structure and motif patterns were related to the unique biological functions of genes [41,42], indicating that in *T. septentrionalis*, HSP70 and HSP90 genes not only shared common functions but also possessed their distinct biological roles. This also highlights the high degree of conservation of HSP70 and HSP90 genes throughout the evolution of *T. septentrionalis*.

The expression profiles of HSP70 and HSP90 genes across different tissues in *T. septentrionalis* revealed that all 12 HSP70 and 4 HSP90 genes were expressed in seven tissues, indicating the important role HSP70 and HSP90 genes played in the biological activities of *T. septentrionalis*, being expressed across most organs. The high expression genes across the seven tissues were *hspa8.1* (liver, intestine, and spleen), *hsp90ab1* (brain, heart, and gills), and *hspa8.2* (muscle), suggesting these genes played a broad role in the life activities of *T. septentrionalis*, with *hspa8.1* and *hsp90ab1* being the highest expression in these tissues. We speculate that these two genes perform indispensable roles, acting as critical hubs in life activities. Furthermore, studies showed that *hspa8* has various cellular functions, mostly through cooperation with chaperones [43], including involvement in protein folding, degradation, and import into organelles. In the liver, the four genes expressed at higher levels were *hspa8.1*, *hsp90ab1*, *hspa5*, and *hsp90b1*. Given the liver’s role as the primary organ for metabolism and immune defense in fish [44], we speculate the primary functions of these four genes pertain to metabolism or immunity. Furthermore, we observed varying expression patterns of the same genes across different tissues, indicating tissue-specific expression of HSP70 and HSP90 genes. For example, the *hspa8.2* gene displays elevated expression levels in muscle tissue, while its expression is notably lower in the heart and spleen, demonstrating a distinct tissue-specific expression profile. Likewise, both zebrafish HSP70 and HSP90 genes during normal development, as well as HSP70 gene expression in turbot under abiotic stress, demonstrated significant tissue-specific expression [2,45]. The expression profiles of HSP70 and HSP90 genes during different developmental stages of *T. septentrionalis* showed that all 16 HSP70 and HSP90 genes were expressed during eight developmental stages, underlining the universal and crucial role of HSP70 and HSP90 genes in the developmental activities of *T. septentrionalis*. For example. In zebrafish, the HSP70 and HSP90 genes play roles in promoting cell growth and development, and additionally, they are involved in facilitating cell differentiation [24,45]. The *hsp90ab1* gene exhibited the highest expression across all eight developmental stages, and among the top five highly expressed genes (*hsp90ab1*, *hspa8.1*, *hspa5*, *hsp90aa1*, *hsp90b1*) during all developmental stages, three were HSP90 genes, highlighting the vital role HSP90 genes played in the developmental activities of *T. septentrionalis*. Studies have demonstrated [24] that HSP90 genes are involved in fundamental cellular processes and regulatory pathways, such as apoptosis and cell cycle control. Additionally, it was observed that the expression profiles of identical genes vary across different developmental stages, suggesting a developmental stage-specific expression pattern for HSP70 and HSP90 genes. For instance, *hsp90b1* displays elevated expression levels in hatch 20-day juveniles compared to the cleavage stage, indicating a pronounced developmental stage-specific expression pattern. Similarly, the HSP70 gene in *Paralichthys olivaceus* and the HSP90 gene in zebrafish exhibit notable specificity across various developmental stages during normal development [46,47]. The expression profiles of HSP70 and HSP90 genes under abiotic stresses in *T. septentrionalis* indicated that under the four studied abiotic stresses, most HSP70 and HSP90 genes were upregulated for expression. This suggested that HSP70 and HSP90 genes in *T. septentrionalis* were broadly involved in environmental stress responses, consistent with studies on Chinese seabass [10]. In the liver, compared to the control group, under high-temperature stress, all genes except *hspa8.2* were upregulated; under high salinity stress, 11 genes were upregulated; under low salinity stress, 10 genes were upregulated; and under ammonia stress, 12 genes were upregulated. In the gills, compared to the control group, under ammonia stress, 10 genes were upregulated. In summary, under high temperature, high salinity, low salinity, and ammonia stress, seven genes were consistently upregulated: *hspa4*, *hspa5*, *hspa9*, *hspa13*, *hspa14*, *hyou1*, and *hsp90b1*, indicating these seven genes were involved in the stress response of *T. septentrionalis* and played crucial roles in responding to abiotic stress. The qPCR validation experiment demonstrated consistent expression trends for the majority of HSP70 and HSP90 genes with the transcriptomic data, affirming the reliability of our transcriptomic findings and this reaffirms the significant roles of HSP70 and HSP90 in *T. septentrionalis*’ response to abiotic stresses.

## 4. Materials and Methods

### 4.1. Identification and Nomenclature of HSP70 and HSP90 Gene Family Members in T. septentrionalis

To identify the HSP70 and HSP90 gene family members in *T. septentrionalis*, first, the genome sequences, protein sequences, and annotation files of *T. septentrionalis* were obtained from our laboratory [15]. Then, to enhance the accuracy of screening gene family members, two methods (BLASTp and HMMER) [48,49] were concurrently employed to query the HSP70 and HSP90 gene family members in *T. septentrionalis*. Initially, the BLASTp program was used for sequence alignment. Relevant HSP70 and HSP90 protein sequences of *H. sapiens*, *M. musculus*, *G. gallus*, *D. rerio*, *O. latipes*, *G. aculeatus*, *L. crocea*, *O. niloticus*, *S. salar*, *I. punctatus*, and *L. oculatus* (Appendix A) published in databases such as NCBI (https://www.ncbi.nlm.nih.gov/, accessed on 10 October 2023), Uniprot (https://www.uniprot.org/, accessed on 10 October 2023), and ENSEMBL (http://ensemblgenomes.org/, accessed on 11 October 2023) were downloaded and used as query sequences. These sequences were aligned against *T. septentrionalis* HSP70 and HSP90 protein sequences using the TBtools software (version 2.092, Dr. Chengjie Chen, Guangdong, China, accessed on 12 October 2023) with a BLASTp comparison (*E*-value ≤ e^−5^). Secondly, the HMMER method was applied for alignment. Hidden Markov models (hmm) of HSP70 (PF00012) and HSP90 (PF00183) genes were downloaded from the Pfam protein database (http://pfam-legacy.xfam.org/, accessed on 13 October 2023) [50], and *T. septentrionalis* protein sequences were searched against these hmms using HMM search in TBtools software (version 2.092, Dr. Chengjie Chen, Guangdong, China, accessed on 13 October 2023). The intersection of BLASTp and HMMER results yielded candidate gene IDs as the initial identification results for HSP70 and HSP90 gene family members. Next, the target sequences of candidate genes were extracted from *T. septentrionalis* protein sequences using TBtools software (version 2.092, Dr. Chengjie Chen, Guangdong, China, accessed on 15 October 2023). Additionally, to further validate the accuracy of the results, the identified gene family members were verified using the search function on the online website HMMER (https://www.ebi.ac.uk/Tools/hmmer/search/, accessed on 15 October 2023) [49]. The HSP70 and HSP90 gene family members in *T. septentrionalis* were named following the nomenclature conventions for teleost fish.

### 4.2. Physicochemical Property Analysis and Phylogenetic Analysis

The molecular characteristics of HSP70 and HSP90 genes, such as molecular weight, isoelectric point, instability index, aliphatic index, and average hydrophilicity, were predicted using the ProtParam tool on the ExPASy website (https://www.expasy.org/, accessed on 16 October 2023) [51].

To elucidate the phylogenetic relationships of the HSP70 and HSP90 gene families across different species, including *T. septentrionalis*, teleosts, and mammals, a phylogenetic tree was constructed using the protein sequences of HSP70 and HSP90 from these groups. All protein sequences required for constructing the evolutionary tree from other species are provided in Appendix A. The Clustal W function within MEGA 11 software [52] was used to perform multiple sequence alignments of the HSP70 and HSP90 protein sequences and those identified in the aforementioned species. The phylogenetic trees were built based on the Neighbor-joining (NJ) method, with bootstrap testing repeated 1000 times and other parameters set to default values. The phylogenetic trees were further optimized and displayed using the online website iTOL (https://itol.embl.de/upload.cgi, accessed on 17 October 2023) [53].

### 4.3. Gene Structure, Chromosome Distribution Analysis, Motif, and Conserved Domain Analysis

To analyze the gene structure of HSP70 and HSP90 in *T. septentrionalis*, the genome annotation file of *T. septentrionalis* was utilized, and TBtools software (version 2.092, Dr. Chengjie Chen, Guangdong, China, accessed on 18 October 2023) was employed to construct gene structure diagrams for HSP70 and HSP90, followed by creating a composite illustration combining gene structure with phylogenetic relationships. Based on the annotation information from the *T. septentrionalis*’s genome, TBtools was also used to draw the chromosomal distribution maps of HSP70 and HSP90 genes, with modifications for gene density on the chromosomal localization diagrams. The MEME Suite online website (https://meme-suite.org/meme/tools/meme, accessed on 18 October 2023) [54] was used to predict and analyze conserved motifs within the protein sequences of *T. septentrionalis*’s HSP70 and HSP90, setting the number of motifs to 20, with other parameters at their default settings. The Batch CD-Search program on the NCBI website was utilized for conserved domain searches within the HSP70 and HSP90 protein sequences. MEGA11 software (version 11, Mega Limited, Auckland, New Zealand, accessed on 19 October 2023) was used to construct the phylogenetic relationships of *T. septentrionalis*’s HSP70 and HSP90 genes, and TBtools software (version 2.092, Dr. Chengjie Chen, Guangdong, China, accessed on 19 October 2023) was used to draw a composite illustration depicting motifs, conserved domains, and phylogenetic relationships.

### 4.4. Protein Structure and Subcellular Localization

The secondary structure of HSP70 and HSP90 proteins was predicted using the online website SOPMA (https://npsa-prabi.ibcp.fr/cgi-bin/npsa_automat.pl?page=/NPSA/npsa_sopma.html, accessed on 21 October 2023) [55]. The tertiary structures of HSP70 and HSP90 proteins were forecasted using the online website SWISS-MODEL (https://swissmodel.expasy.org/, accessed on 22 October 2023) [56], and the quality of the obtained models was assessed using SAVES v.6.0 (https://saves.mbi.ucla.edu/, accessed on 22 October 2023). Subcellular localization of HSP70 and HSP90 genes was performed using the WoLF PSORT website (https://psort.hgc.jp/form2.html, accessed on 22 October 2023) [57].

### 4.5. Signal Peptides Prediction, Transmembrane Structural Domain Prediction and Selection Test of Duplicated HSP70 and HSP90 Genes

Signal peptides of HSP70 and HSP90 proteins were predicted using the online website SignalP 5.0 (https://services.healthtech.dtu.dk/services/SignalP-5.0/, accessed on 23 October 2023). Transmembrane domains of HSP70 and HSP90 proteins were forecasted using the online website TMHMM 2.0 (https://services.healthtech.dtu.dk/services/TMHMM-2.0/, accessed on 23 October 2023). The Simple Ka/Ks Calculator function of TBtools software (version 2.092, Dr. Chengjie Chen, Guangdong, China) was utilized to calculate the nonsynonymous (Ka), synonymous (Ks), and Ka/Ks ratio for duplicated pairs of HSP70 and HSP90 genes.

### 4.6. Expression Profiles of HSP70 and HSP90 Genes in Tissues of T. septentrionalis

The specimens of *T. septentrionalis* for this study were all sourced from Tianyuan Aquatics Co., Ltd., Yantai City, Shandong Province, China.

One-year-old *T. septentrionalis*es were selected for tissue sampling, which included seven types of tissues: heart, brain, liver, spleen, gills, intestine, and muscle. The seawater temperature for culturing *T. septentrionalis* was maintained at 20~22 °C. Five one-year-old *T. septentrionalis* were randomly sampled and anesthetized with 50 mg/L MS-222. The entire sampling process was conducted on an ice tray within a sterile operating bench. After dissection, the relevant tissues were rapidly rinsed with 0.9% saline solution, placed into 2 mL cryovials, and immediately stored in liquid nitrogen before being transferred to a −80 °C ultra-low temperature freezer for subsequent RNA-seq transcriptome analysis. The transcriptome library and analysis process are as follows: First, RNA was extracted from tissues using standard extraction methods, and then RNA integrity was evaluated using the RNA Nano 6000 Assay Kit with the Bioanalyzer 2100 system (Agilent Technologies, Santa Clara, CA, USA). Second, library construction was performed using the NEB standard library preparation method. Third, sequencing was carried out on the Illumina Novaseq platform. Fourth, raw sequencing data underwent quality control. Fifth, Clean Reads were aligned to the reference genome using HISAT2 software (version 2.1.0, CCB at JHU, Baltimore, MD, USA) to obtain the positional information of Reads on the reference genome. Additionally, FPKM, which represents the number of Fragments mapped to a gene, divided by the total length of gene exons and then divided by the total length of all genes, was calculated. RNA quality validation charts have been presented in Appendix A.

In the transcriptome result files of seven tissues from *T. septentrionalis*, the FPKM values for HSP70 and HSP90 gene expressions were retrieved and transformed using log_2_ (FPKM + 1). TBtools software (version 2.092, Dr. Chengjie Chen, Guangdong, China) was utilized to create a heatmap to study the expression profiles of HSP70 and HSP90 Genes across different tissues in *T. septentrionalis*.

### 4.7. Expression Profiles of HSP70 and HSP90 Genes during Different Developmental Stages of T. septentrionalis

Broodstock in gonadal development stage IV or above was selected for spawning, and eggs were collected for microscopic examination and sampling. The developmental staging of *T. septentrionalis* referred to the study by Guan et al. [58], including the cleavage stage, blastula stage, gastrula stage, neuroembryonic stage, tail bud stage, and muscle effect stage. After filtration through a 60-mesh sieve [59], 30 eggs were collected, and 15 specimens were taken for both newly hatched larvae (Dph0) and 20-day-old juveniles (Dph20). All samples were quickly placed into 2 mL cryovials and then stored in liquid nitrogen before being transferred to a −80 °C ultra-low freezer for subsequent RNA-seq transcriptome analysis.

In the transcriptome result files of *T. septentrionalis* across different developmental stages, FPKM values for the expression of HSP70 and HSP90 genes were retrieved and transformed using log_2_ (FPKM + 1). TBtools software (version 2.092, Dr. Chengjie Chen, Guangdong, China) was utilized to create a heatmap to investigate the expression profiles of HSP70 and HSP90 Genes during various developmental stages of *T. septentrionalis*.

### 4.8. Expression Profiles of HSP70 and HSP90 Genes under Abiotic Stresses

The abiotic stress experiments included high temperature, high salinity, low salinity, and ammonia nitrogen stress, with the subjects being healthy and vigorous *T. septentrionalis* adults of similar size (body length 20 ± 0.5 cm, body weight 120 ± 0.5 g). Initially, one high-temperature group (temperature 30 °C), one high-salinity group (40‰), one low-salinity group (15‰), one ammonia nitrogen group (10.60 mg/L), and one blank control group (ambient seawater salinity 30‰, temperature 20 °C, ammonia concentration 0 mg/L) were set up. Subsequently, in the high-temperature and ammonia experiments, subjects acclimated for one week were directly placed into the experimental groups corresponding to the designed stress environments. For the high-salinity and low-salinity experiments, subjects acclimated for one week and were gradually adjusted to the stress environments corresponding to the experimental design, undergoing a 24 h stress culture experiment. During the experiment, feed was administered twice daily at 08:00 and 17:00, with leftover feed removed after feeding. Daily bottom suction and water changes were conducted, along with continuous aeration. Each experimental group had three replicates, with each replicate containing 10 fish.

All samples were collected at the 24th hour of the stress experiment. For the high temperature, high salinity, and low salinity stress experiments, the liver of *T. septentrionalis* was sampled, while for the ammonia nitrogen stress experiment, both the liver and gills were sampled. In each group, three experimental fish were randomly selected and anesthetized with 50 mg/L MS-222. The entire sampling process was conducted on an ice tray within a sterile operating bench. After dissection, the relevant tissues were quickly rinsed with 0.9% saline solution, placed into 2 mL cryovials, and immediately stored in liquid nitrogen before being transferred to a −80 °C ultra-low freezer for subsequent transcriptome analysis.

FPKM values for the expression of HSP70 and HSP90 genes under abiotic stress were retrieved from the transcriptome result files and transformed using log10 (FPKM + 1). TBtools software (version 2.092, Dr. Chengjie Chen, Guangdong, China) was utilized to create a heatmap to investigate the expression profiles of HSP70 and HSP90 Genes under different environmental stresses. The heatmap data for different tissues, developmental stages, and abiotic stresses are provided in Appendix A.

### 4.9. Quantitative Real-Time PCR (qPCR)

To further validate the expression levels of HSP70 and HSP90 genes in *T. septentrionalis* under environmental stress, qPCR was performed. Total RNA from the liver of *T. septentrionalis* subjected to environmental stress was extracted using the TIANGEN Total RNA Extraction Kit, following the manufacturer’s instructions. The concentration and quality of the extracted total RNA were assessed using a NanoDrop spectrophotometer and agarose gel electrophoresis, respectively. cDNA was synthesized using the HiScript III RT SuperMix for qPCR (+gDNA wiper) (Vazyme) kit. Primers for the HSP70 and HSP90 genes and the internal reference gene β-actin were designed using the Primer-BLAST program on the NCBI website, with all primer sequences provided in Appendix A. qRT-PCR was conducted using the ChamQ SYBR Color qPCR Master Mix (Vazyme) to detect the expression of HSP70 and HSP90 genes in *T. septentrionalis* in response to different environmental stresses. The 20 μL reaction system included 10 μL of 2 × ChamQ SYBR Color qPCR Master Mix, 0.4 μL of forward primer, 0.4 μL of reverse primer, 0.4 μL of 50 × ROX Reference Dye 1, 2 μL of template, and 6.8 μL of ddH_2_O. The PCR program was as follows: pre-denaturation at 95 °C for 30 s; 40 cycles of denaturation at 95 °C for 10 s and annealing at 60 °C for 30 s, followed by a melting curve analysis: 95 °C for 15 s, 60 °C for 60 s, 95 °C for 15 s. The relative expression levels of the HSP70 and HSP90 genes were calculated using the 2^−ΔΔCT^ method [60]. Statistical analysis of the relative mRNA expression level between transcriptomic data and qPCR data was performed using Student’s t-test with SPSS 26.0 (v26) software, and the differences were considered significant when the *p* value < 0.05 (*) and extremely significant when the *p* value < 0.01 (**).

## 5. Conclusions

In this study, we identified 12 HSP70 genes (*hsp70*, *hspa1b*, *hspa4*, *hspa4a*, *hspa4l*, *hspa5*, *hspa8.1*, *hspa8.2*, *hspa9*, *hspa13*, *hspa14*, and *hyou1*) and 4 HSP90 genes (*hsp90aa1*, *hsp90ab1*, *hsp90b1*, and *trap1*) in *T. septentrionalis*. This study employed bioinformatics methods to conduct a series of analyses of the HSP70 and HSP90 gene families in *T. septentrionalis*, encompassing phylogenetic analysis, motif analysis, conserved domain analysis, and chromosomal localization. Additionally, a thorough investigation of HSP70 and HSP90 gene expression profiles was conducted across different tissues, developmental stages, and under various abiotic stress conditions in *T. septentrionalis*, elucidating their roles in normal growth and development processes as well as their responses to abiotic stress. To further validate the accuracy of the transcriptome results, qPCR analysis was performed on the HSP70 and HSP90 genes under three abiotic stress conditions. This research provides fundamental insights into the molecular regulatory mechanisms of the HSP70 and HSP90 gene families in *T. septentrionalis* under environmental stresses. The identified genes can serve as reference markers for detecting environmental stress in organisms, thereby facilitating the optimization of aquaculture techniques for *T. septentrionalis*. Building on these findings, future research endeavors will include additional experimental components, such as cellular localization studies, to further explore the functional roles of HSP gene families and validate the in silico data obtained in this study.

## Figures and Tables

**Figure 1 ijms-25-05706-f001:**
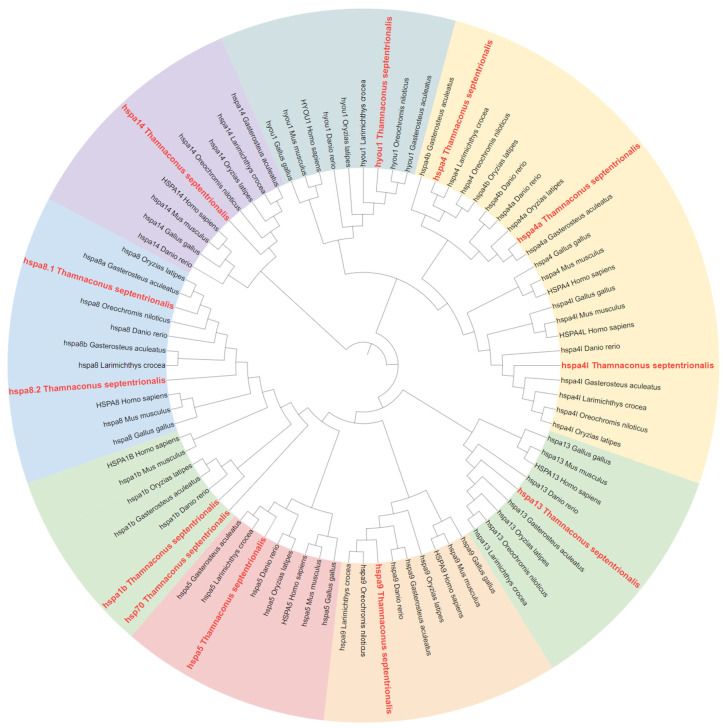
Phylogenetic tree of the HSP70 gene family from *T. septentrionalis* and other species (total 9 species). The phylogenetic tree was constructed by the MEGA 11 using the neighbor-joining method, and the statistical robustness of the tree was estimated by bootstrapping with 1000 replicates. The HSP70 genes of *T. septentrionalis* are marked in red, and its HSP70 gene family is divided into eight groups (hspa1, hspa4, hspa5, hspa8, hspa9, hspa13, hspa14, and hyou1), which are marked with different colors.

**Figure 2 ijms-25-05706-f002:**
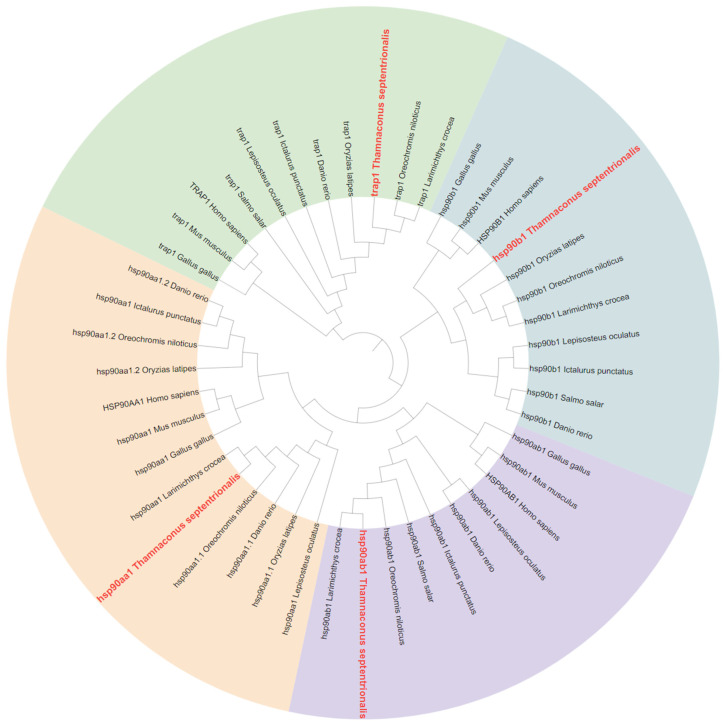
Phylogenetic tree of the HSP90 gene family from *T. septentrionalis* and other species (total 11 species). The HSP90 genes of *T. septentrionalis* are marked in red, and its HSP90 gene family is divided into four groups (hsp90aa1, hsp90ab1, hsp90b1, and trap1), which are marked with different colors.

**Figure 3 ijms-25-05706-f003:**
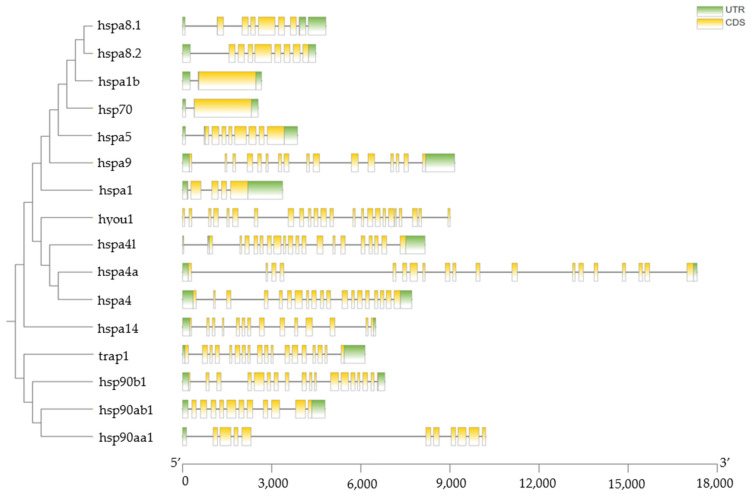
The combination of phylogenetic tree and gene structure. The left side of the figure shows the phylogenetic tree of the HSP70 and HSP90 genes of *T. septentrionalis*, and the right side shows the gene structures and gene lengths.

**Figure 4 ijms-25-05706-f004:**
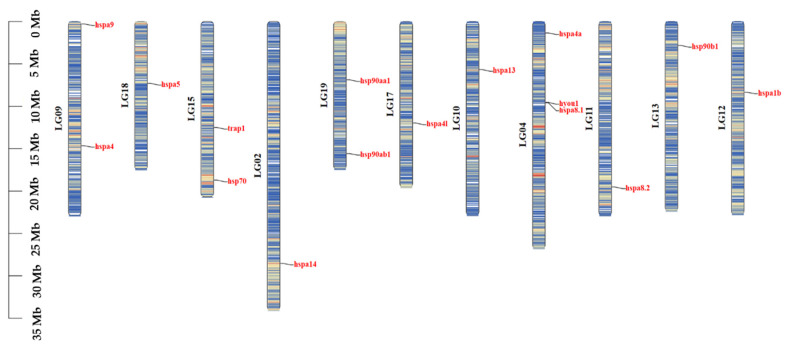
Chromosomal distribution of HSP70 and HSP90 genes in *T. septentrionalis*. Genes and chromosomes are labeled in red and black, respectively.

**Figure 5 ijms-25-05706-f005:**
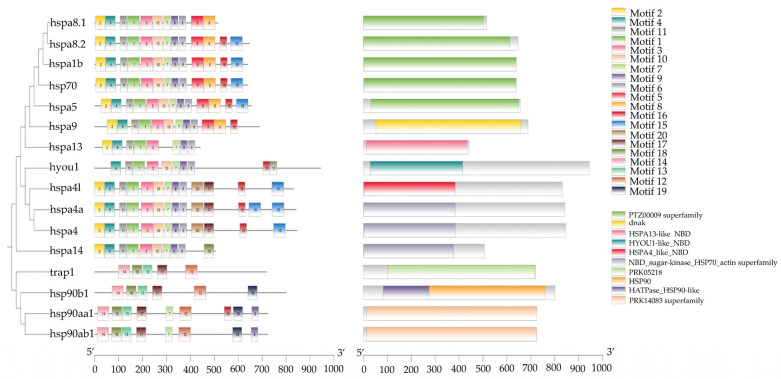
The combination of phylogenetic tree and motifs and conserved domains. The left side of the figure shows the phylogenetic tree of the HSP70 and HSP90 genes of *T. septentrionalis*, the middle shows the distribution of the motifs and each colored rectangle represents a motif, the right side shows the identification of conserved domains and each colored rectangle represents a conserved structural domain.

**Figure 6 ijms-25-05706-f006:**
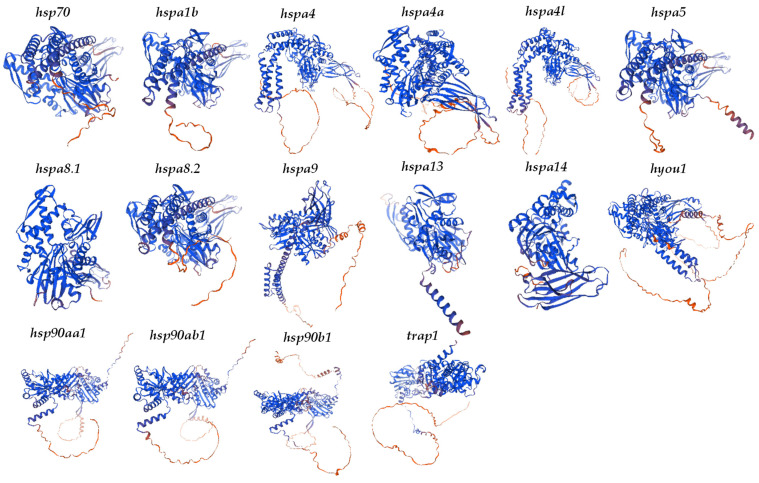
Three-dimensional structural projection of HSP70 and HSP90 proteins. Blue parts represent α helix and red parts represent β Turn.

**Figure 7 ijms-25-05706-f007:**
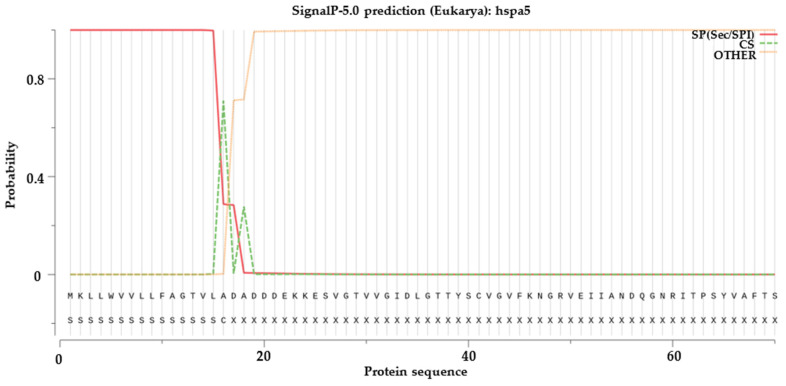
Signal peptide prediction analysis of HSP70 protein (*hspa5* and *hyou1*) and HSP90 protein (*hsp90b1*).

**Figure 8 ijms-25-05706-f008:**
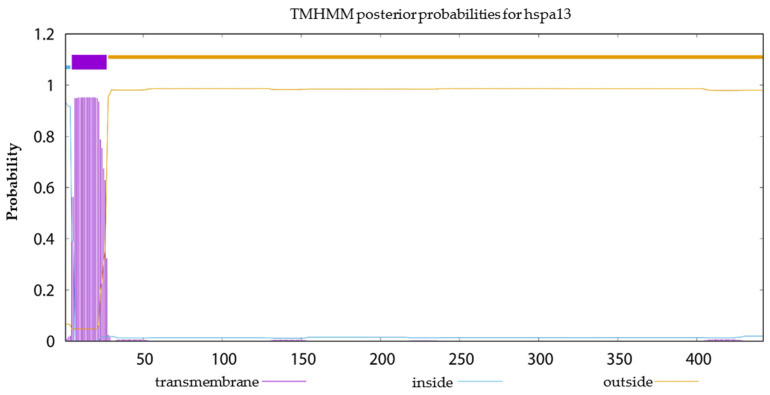
Prediction of protein transmembrane structural domains of HSP70 protein (*hspa13*).

**Figure 9 ijms-25-05706-f009:**
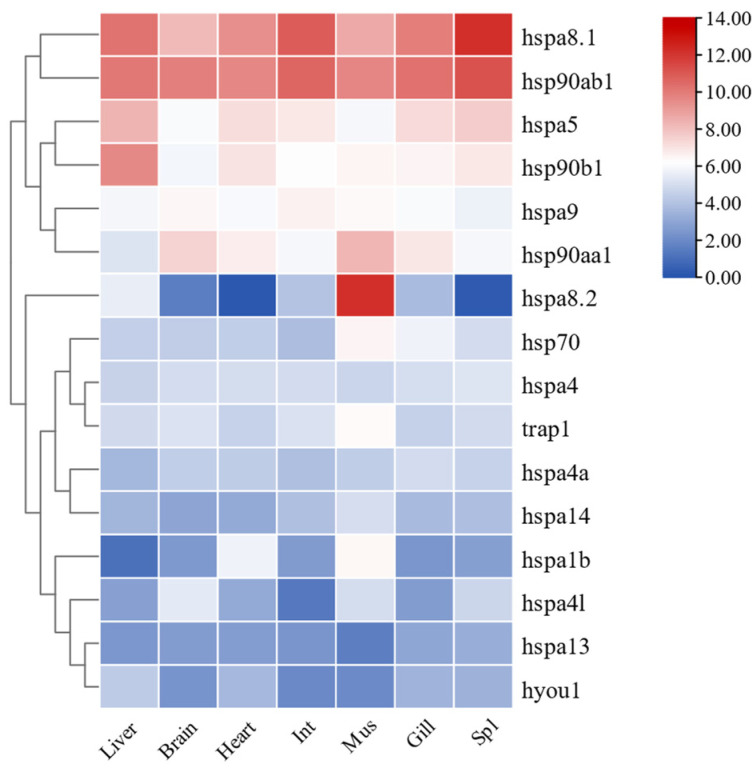
Expression profiles of HSP70 and HSP90 genes in different tissues of *T. septentrionalis*. Cells with different colors correspond to different expression levels, which were normalized into log_2_ (FPKM + 1). Int, Mus, and Spl represent the intestine, muscle, and spleen, respectively.

**Figure 10 ijms-25-05706-f010:**
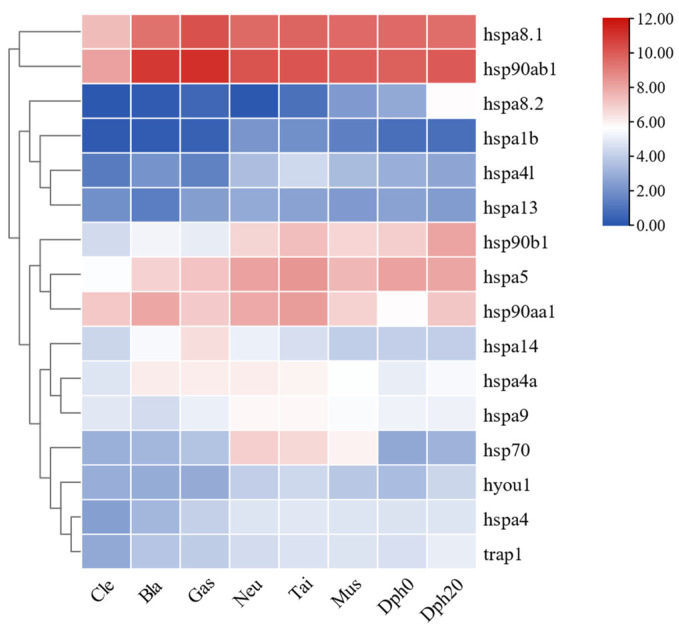
Expression profiles of HSP70 and HSP90 genes in different developmental periods of *T. septentrionalis*. Cells with different colors correspond to different expression levels, which were normalized into log_2_ (FPKM + 1). Cle, Bla, Gas, Neu, Tai, Mus, Dph0, and Dph20 represent cleavage stage, blastocyst stage, gastrula stage, neuroembryonic stage, tail bud stage, muscle effect stage, hatch 0-day larvae, and hatch 20 days juvenile, respectively.

**Figure 11 ijms-25-05706-f011:**
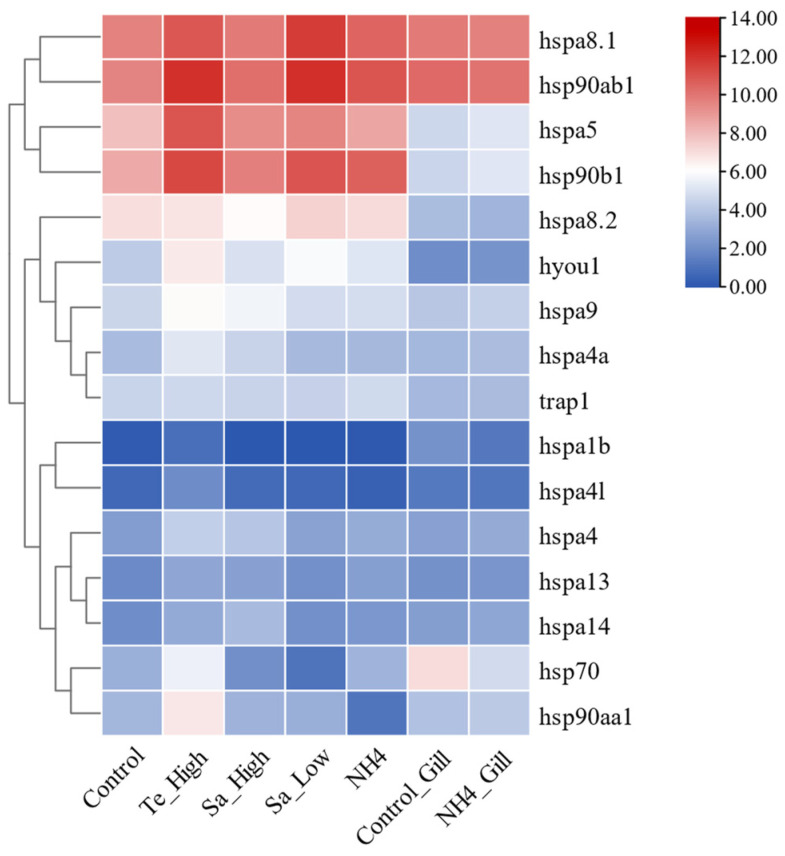
Expression profiles of HSP70 and HSP90 genes in *T. septentrionalis* under different abiotic stresses (high temperature, high salt, low salt, and ammonia stresses). Cells with different colors correspond to different expression levels, which were normalized into log_2_ (FPKM + 1). The tissue used in Panel Control, Te_High, Sa_High, and Sa_Low was the liver, The tissue sampled in Panel Control_Gill and NH4_Gill was the gill. Te_High, Sa_High, Sa_Low, and NH4 denote high temperature 30 °C, high Salt 40‰, Low salt 15‰, and ammonia nitrogen 10.60 mg/L, respectively.

**Figure 12 ijms-25-05706-f012:**
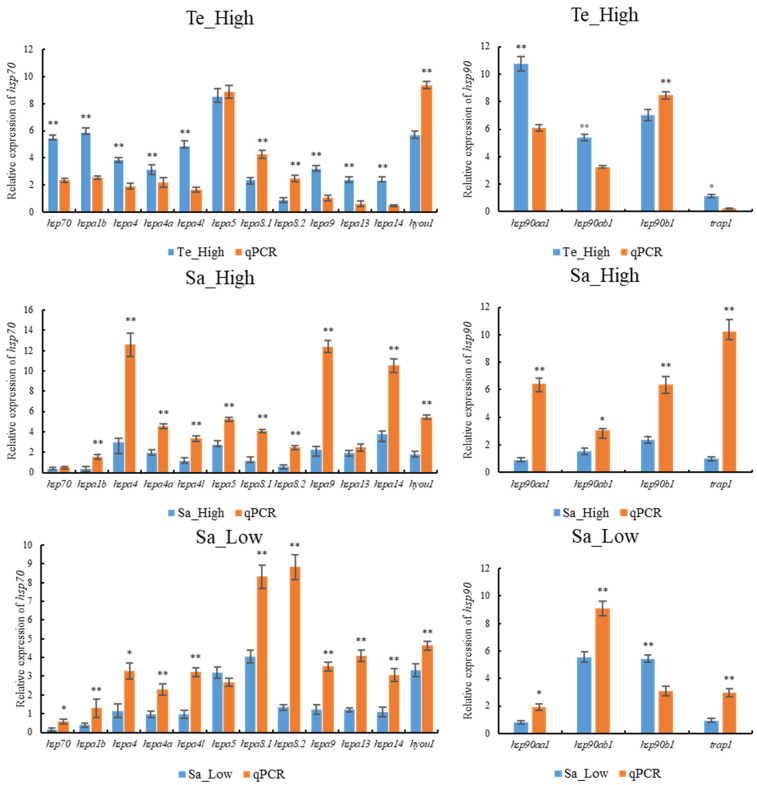
Validation of HSP70 and HSP90 gene expression under different environmental stresses by Quantitative Real-time PCR (qPCR). The tissue used in qPCR was the liver. The left side of the figure shows the relative expression of the HSP70 genes under environmental stress, while the right side of the figure shows the HSP90 genes. Blue bars indicate transcriptome results, and orange bars indicate qPCR results. Te_High, Sa_High, and Sa_Low denote high temperature 30 °C, high Salt 40‰, and Low salt 15‰, respectively. The mRNA expression levels were determined by qPCR analysis using the 2^−ΔΔCt^ method. * and ** indicate the significant differences at *p* < 0.05 and *p* < 0.01 between qPCR results and transcriptome results for the same gene, respectively.

**Table 1 ijms-25-05706-t001:** Summary of sequence characteristics of HSP70 and HSP90 genes in *T. septentrionalis*.

Gene Name	NCBI Accession Number	Number of Amino Acid	Molecular Weight (Da)	Theoretical pI	Instability Index	Aliphatic Index	Grand Average of Hydropathicity
*hsp70*	PP349927	639	70,266.65	5.44	36.44	84.98	−0.423
*hspa1b*	PP349926	639	70,106.24	5.53	37.52	83.93	−0.412
*hspa4*	PP349928	846	95,013.44	5.04	43.03	72.97	−0.628
*hspa4a*	PP349923	842	94,144.84	5.17	43.54	76.28	−0.534
*hspa4l*	PP349922	832	92,954.52	5.46	41.20	79.68	−0.487
*hspa5*	PP349929	656	72,413.78	4.97	30.55	84.39	−0.482
*hspa8.1*	PP357443	515	56,702.91	5.90	36.45	81.98	−0.405
*hspa8.2*	PP349925	647	70,877.18	5.27	39.45	81.82	−0.422
*hspa9*	PP349930	688	74,170.99	6.24	45.59	81.79	−0.374
*hspa13*	PP339453	442	47,831.97	5.52	40.48	102.15	0.082
*hspa14*	PP349924	506	54,814.52	5.66	38.49	94.43	−0.052
*hyou1*	PP349931	945	106,412.40	6.03	45.14	79.47	−0.527
*hsp90aa1*	PP349934	725	83,238.60	5.06	35.70	82.81	−0.625
*hsp90ab1*	PP349932	724	83,206.20	4.87	39.91	82.56	−0.640
*hsp90b1*	PP349933	801	91,991.32	4.74	38.96	79.38	−0.693
*trap1*	PP349935	719	81,840.08	5.99	45.54	83.99	−0.440

**Table 2 ijms-25-05706-t002:** Secondary structure prediction and subcellular location prediction of HSP70 and HSP90 proteins.

Gene Name	α Helix	Extended Strand	β Turn	Random Coil	Subcellular Location	Three-Dimensional Structural Quality Parameters
*hsp70*	42.10%	18.47%	5.95%	33.49%	Nucleus	94.0%
*hspa1b*	42.57%	18.94%	7.20%	31.30%	Cytosol	94.5%
*hspa4*	42.55%	14.18%	3.07%	40.19%	Cytosol	90.6%
*hspa4a*	43.11%	14.01%	3.09%	39.79%	Cytosol	92.2%
*hspa4l*	43.63%	14.06%	3.25%	39.06%	Cytosol	91.9%
*hspa5*	43.75%	18.60%	6.71%	30.95%	Endoplasmic reticulum	93.6%
*hspa8.1*	33.79%	23.30%	8.54%	34.37%	Cytosol	93.2%
*hspa8.2*	41.58%	17.77%	6.96%	33.69%	Cytosol	93.8%
*hspa9*	43.75%	19.77%	8.14%	28.34%	mitochondrion	92.9%
*hspa13*	40.27%	21.27%	5.66%	32.81%	Endoplasmic reticulum	93.7%
*hspa14*	34.78%	24.90%	5.14%	35.18%	Cytosol	93.6%
*hyou1*	48.99%	13.23%	3.28%	34.50%	Endoplasmic reticulum	89.9%
*hsp90aa1*	52.55%	13.66%	5.38%	28.41%	Cytosol	89.5%
*hsp90ab1*	50.14%	13.54%	4.56%	31.77%	Cytosol	90.4%
*hsp90b1*	54.68%	12.36%	3.75%	29.21%	Endoplasmic reticulum	88.2%
*trap1*	45.20%	13.77%	5.70%	35.33%	mitochondrion	88.8%

**Table 3 ijms-25-05706-t003:** Ka, Ks, and Ka/Ks * ratios of duplicated HSP70 and HSP90 gene pairs.

Gene Pair	Ka	Ks	Ka/Ks
*hsp70-hspa1b*	0.104125523	0.918427	0.113374
*hspa4-hspa4a*	0.175276849	1.508566	0.116188
*hspa4-hspa4l*	0.307067194	1.683617	0.182385
*hspa4a-hspa4l*	0.285118286	1.702043	0.167515
*hspa8.1-hspa8.2*	0.06700074	1.461276	0.045851
*hsp90aa1-hsp90ab1*	0.116635393	1.012976	0.115141

* Ka: nonsynonymous substitution rate; Ks: synonymous substitution rate.

## Data Availability

The datasets analyzed for this study can be found in NCBI: https://www.ncbi.nlm.nih.gov/ (accessed on 21 May 2024). The accession numbers can be found below: SRR27393461~SRR27393490 and SRR28205501~SRR28205545.

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
