# Peer review of "Genome-Wide Identification, Molecular Characterization, and Expression Analysis of the HSP70 and HSP90 Gene Families in Thamnaconus septentrionalis"

_ijms, 2024, doi:10.3390/ijms25115706_

Round 1

Reviewer 1 Report

Comments and Suggestions for Authors

In the entire manuscript, the authors have presented various in silico information regarding conserved domain analysis, protein structure determination, and cellular localization.

1.I will suggest to provide more extensive information on these, especially the cellular localization of HSP proteins obtained from wet lab experiments.

2.In figure 6: The three-dimensional structure of HSP70 and HSP90 proteins was given, but no quality parameters was shown in table 2.

3.In Figure 12. Validation of HSP70 and HSP90 gene expression through qPCR, plz provide the standard error of mean in bars.

4. Kindly specify the tissue used for the expression analysis of HSPs under abiotic stress conditions.

5. It will be more better, if the authors could provide more information on tissue specific and development stage specific expression and validation of HSP genes under abiotic stress, to explain their functional significance.

Author Response

Dear reviewer,

Thank you very much for taking the time to review our manuscript titled " Genome-Wide Identification, Molecular Characterization, and Expression Analysis of the HSP70 and HSP90 gene families in Thamnaconus septentrionalis" submitted to International Journal of Molecular Sciences. We sincerely appreciate your thoughtful comments and valuable feedback, which have greatly contributed to the improvement of our work.

We have carefully considered each of your suggestions and have made the necessary revisions to address the concerns raised. In the revised manuscript, we have provided additional clarification, updated data, and made appropriate changes to enhance the overall quality and clarity of the manuscript.

A detailed response to each of your comments and queries is provided below:

  1. I will suggest to provide more extensive information on these, especially the cellular localization of HSP proteins obtained from wet lab experiments.

Thank you for your suggestion, this study aims to conduct a preliminary genome-based mining and characterization of HSPs genes in Thamnaconus septentrionalis. Based on the results of this study, we intend to incorporate additional experiments, including cellular localization in a subsequent study to further investigate the function of the HSPs gene family and to validate the data from this study. We have added this description in the Conclusions part.

  1. In figure 6: The three-dimensional structure of HSP70 and HSP90 proteins was given, but no quality parameters was shown in table 2.

Thank you for your suggestion. We have included the Three-dimensional structural quality parameters in Table 2, and the images displaying the results of quality parameters are presented in the supplementary materials Table S5.

  1. In Figure 12. Validation of HSP70 and HSP90 gene expression through qPCR, plz provide the standard error of mean in bars.

Thank you for your suggestion, we have added the standard error of mean to Figure 12.

  1. Kindly specify the tissue used for the expression analysis of HSPs under abiotic stress conditions.

Thank you for your suggestion. We have added the tissue names to the caption of Figure 12.

  1. It will be more better, if the authors could provide more information on tissue specific and development stage specific expression and validation of HSP genes under abiotic stress, to explain their functional significance.

Thank you for your suggestion. We have incorporated information regarding tissue-specific and developmental stage-specific expression, as well as qPCR validation, into 3 Discussion section.

  1. “ and * ”: What next?

Thank you for your suggestion, we have adjusted the position of 'and'.

  1. how does these in silico information are linked to their expression analysis?

Thank you for your suggestion. We have revised the original description. Please refer to the highlighted section in Section 5 Conclusions for details.

We hope that the revisions meet with your approval. Please do not hesitate to contact us if you require any further information or clarification.

Once again, we would like to express our gratitude for your time and effort in reviewing our manuscript.

Sincerely,

Li Bian and Siqing Chen

Reviewer 2 Report

Comments and Suggestions for Authors

The Authors here present a study conducted to identify HSP70 and HSP90 genes in T. septentrionalis, since  analysis of these gene families was missing. They also comprehensively analysed phylogenetics, gene structure,  chromosomal localization, and expression profiling. The paper is interesting as it provides data for understanding the role of HSP70 and HSP90 623 gene families in T. septentrionalis in response to environmental stresses. In addition, the Authors suggest that the identified genes can serve as molecular markers for early detection of environmental stress, aiding in the optimization of T. septentrionalis aquaculture techniques.

The manuscript is generally well written and results are presented clearly. Nevertheless, this work has a major flaw as statistical analysis and replicates experiments are missing, which provide only weak support to the conclusions and compromise the reliability of the results.

Specific comments.

Review carefully the formatting, e.g. for figure 3 legend and lines 193-202

Page 1, line 40: rephrase the sentence to avoid repetitions (aquatic animals/aquatic organisms).

Page 3, Line 112: according to M&M section this data was not obtained within the same analysis, I suggest to move that info to the following paragraph, also considering that in the following section, the Authors give additional details on this feature.

Page 4, Line 131-132 and 142-143: species names must be in italics.

Figure 5: the text in the figure is not legible. The authors have to find a way to increase the font size to improve the readability.

Page 10, line 254: no statistical analysis has been presented to support this assertion.

Figure 7: improve the readability of the text in the figure.

Figure 9. The Authors should provide additional information to explain how gene expression has been expressed.

Figure 12: No replicates nor statistical analysis have been reported for the validation of gene expression under different environmental stresses.

Page , line 373: where does the evidence for the distribution between the mitochondrion and the nucleus come from? Where these data are reported?

Page 17, line 403: again, on which basis do the Authors claim for significance?

Page 17, line 435-438: The authors should mention what is known about the role of  HSP70 and HSP90 during  fish development.

Page 17, line 446: As stated it seems that HSP70 and HSP90 are expressed in T. septentrionalis only under abiotic stresses. Is that the case?

Page 18, line 456: since the stress was induced by an abiotic factor it is not correct to advocate an immune response.

Methods, line 544: the authors should provide information about the methodology adopted for RNA- seq transcriptome analysis. The Authors might provide a figure showing RNA quality on agarose gel electrophoresis (as supplementary information).

Author Response

Dear reviewer,

Thank you very much for taking the time to review our manuscript titled " Genome-Wide Identification, Molecular Characterization, and Expression Analysis of the HSP70 and HSP90 gene families in Thamnaconus septentrionalis" submitted to International Journal of Molecular Sciences. We sincerely appreciate your thoughtful comments and valuable feedback, which have greatly contributed to the improvement of our work.

We have carefully considered each of your suggestions and have made the necessary revisions to address the concerns raised. In the revised manuscript, we have provided additional clarification, updated data, and made appropriate changes to enhance the overall quality and clarity of the manuscript.

A detailed response to each of your comments and queries is provided below:

  1. Review carefully the formatting, e.g. for figure 3 legend and lines 193-202.

Thank you for your suggestions, we have again checked the formatting of the full manuscript.

  1. Page 1, line 40: rephrase the sentence to avoid repetitions (aquatic animals/aquatic organisms).

Thank you for your suggestion, we have reworked the description of the sentence.

  1. Page 3, Line 112: according to M&M section this data was not obtained within the same analysis, I suggest to move that info to the following paragraph, also considering that in the following section, the Authors give additional details on this feature.

Thank you for your suggestion. Except for chromosomal location, the data in Table 1 are all from the same analysis. We have removed the contents of the Chr column in the table and the description related to chromosomal location in Section 2.1 Identification and Characteristics of HSP70 and HSP90 Genes in T. septentrionalis, leaving only the description of chromosomal location in Section 2.3. Gene Structure and Chromosome Distribution Analysis.

  1. Page 4, Line 131-132 and 142-143: species names must be in italics.

Thank you for your suggestion. We have italicized all species names.

  1. Figure 5: the text in the figure is not legible. The authors have to find a way to increase the font size to improve the readability.

Thank you for your suggestion. We have revised the textual part of Figure 5.

6.Page 10, line 254: no statistical analysis has been presented to support this assertion.

Thank you for your suggestion. We mistakenly used 'significantly' here, which does not imply statistical significance with values less than 1.0, but rather to indicate that their values are less than 1.0. We have rephrased this section accordingly.

7.Figure 7: improve the readability of the text in the figure.

Thank you for your suggestion. We have revised the textual content in Figure 7 accordingly.

  1. Figure 9. The Authors should provide additional information to explain how gene expression has been expressed.

Thank you for your suggestion. We have added descriptions of expression levels (log2(FPKM+1)) to all heatmap captions.

  1. Figure 12: No replicates nor statistical analysis have been reported for the validation of gene expression under different environmental stresses.

Thank you for your suggestion. We have added relevant information on data analysis in Section 4.9. Quantitative Real-Time PCR (qPCR), conducted t-tests (**), and added standard error bars to Figure 12, denoted by * and **.

  1. Page17 , line 373: where does the evidence for the distribution between the mitochondrion and the nucleus come from? Where these data are reported?

Thank you for your suggestion. These data were mentioned in Section 2.5. Protein Structure Prediction and Subcellular Localization and Table 2. Now, we have added the data source location (Table 2) to this sentence.

  1. Page 17, line 403: again, on which basis do the Authors claim for significance?

Thank you for your suggestion. We have once again misused the term 'significantly'. We have revised the description accordingly.

  1. Page 17, line 435-438: The authors should mention what is known about the role of HSP70 and HSP90 during fish development.

Thank you for your suggestion. We have incorporated descriptions of the functions of HSP70 and HSP90 during fish larval development in this section, along with relevant literature references.

  1. Page 17, line 446: As stated it seems that HSP70 and HSP90 are expressed in T. septentrionalisonly under abiotic stresses. Is that the case?

Thank you for your suggestion. HSP70 and HSP90 are also expressed under normal conditions, as detailed in the sections on tissue expression and expression at different developmental stages. ‘Under environmental stress, most of the HSP70 and HSP90 genes show upregulated expression.’ We have revised this description accordingly.

  1. Page 18, line 456: since the stress was induced by an abiotic factor it is not correct to advocate an immune response.

    Thank you for your suggestion, and we have revised this portion of the description.

  1. Methods, line 544: the authors should provide information about the methodology adopted for RNA-seq transcriptome analysis. The Authors might provide a figure showing RNA quality on agarose gel electrophoresis (as supplementary information).

Thank you for your suggestion. We have included information on the transcriptome library construction and analysis. Due to variations in RNA quality assessment methods among different sequencing companies, the results obtained from our detection method are represented in RNA quality validation charts. These charts are supplemented in Excel format in the supplementary materials Table S5.

We hope that the revisions meet with your approval. Please do not hesitate to contact us if you require any further information or clarification.

Once again, we would like to express our gratitude for your time and effort in reviewing our manuscript.

Sincerely,

Li Bian and Siqing Chen

Round 2

Reviewer 1 Report

Comments and Suggestions for Authors
What does the value mean for three-dimensional structural quality parameters? RMSD value or Ramachandran plot?

Author Response

Thank you for the review. The three-dimensional structural quality parameters derived from the Ramachandran plot were above 90%, indicating that the constructed 3D models of the proteins were of high quality. Ramachandran plots are displayed in Supplementary Material Table S4. The relevant description in the article is marked yellow on lines 237-239.

Reviewer 2 Report

Comments and Suggestions for Authors

The Authors answered satisfactorily to the points raised by this reviewer. There is only a minor modification required:

Lines 525-528: check the punctuation  

Author Response

Thank you for your suggestion. The punctuation has been changed and highlighted in yellow.